# MFSR: MeanFlow Distillation for One Step Real-World Image Super Resolution

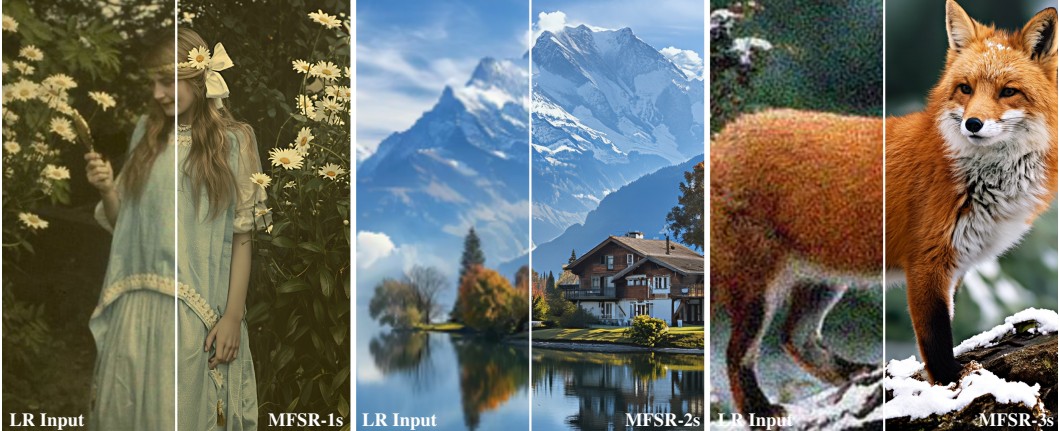

Figure 1: We present MFSR, a high-capacity one- or few-step super-resolution model that delivers photorealistic restoration of real-world low-resolution images. The number of diffusion inference steps is indicated by 's'.

## Abstract

Diffusion- and flow-based models have advanced real-world image super-resolution (Real-ISR), but their multi-step sampling makes inference slow and hard to deploy. One-step distillation alleviates the cost, yet often degrades restoration quality and removes the option to refine with more steps. We present **Mean Flows for Super-Resolution (MFSR)**, a new distillation framework that produces photorealistic, high-fidelity results in a single step while still allowing an optional few-step path for further improvement. Our approach uses *MeanFlow* as the learning target, enabling the student to approximate the mean velocity between arbitrary states of the Probability Flow ODE (PF-ODE) and effectively capture the teacher's dynamics without explicit rollouts. To better leverage pretrained generative priors, we additionally improve original MeanFlow's Classifier-Free Guidance (CFG) formulation with teacher CFG distillation strategy, which enhances restoration capability and preserves fine details. Experiments on both synthetic and real-world benchmarks demonstrate that MFSR achieves efficient, flexible, and high-quality super-resolution, delivering results on par with or even better than multi-step teachers while requiring much lower computational cost.

## 1 Introduction

Image Super-Resolution (ISR) (Dong et al., 2014; Kim et al., 2016; Ledig et al., 2017) aims to reconstruct High-Resolution (HR) image from Low-Resolution (LR) inputs. Traditional ISR methods typically downsample HR images to form training pairs. However, such approaches fall short when dealing with real-world images degraded by complex and unknown processes. Recent research has shifted toward Real-World ISR (Real-ISR) (Zhang et al., 2021; Wang et al., 2021), a more challenging yet practically valuable setting.

Early progress in Real-ISR was largely driven by Generative Adversarial Networks (GANs) (Goodfellow et al., 2014; Mirza, 2014), where adversarial training encouraged sharper textures and per-

ceptual realism. Despite their success, GAN-based methods often suffer from unstable training and tend to introduce artifacts. This has motivated exploration of more powerful generative paradigms.

Recently, diffusion- and flow-based generative models (Song et al., 2020; Ho et al., 2020; Rombach et al., 2022; Liu et al., 2022; Liu, 2022; Lipman et al., 2022; Albergo et al., 2023) have shown superior image generation capabilities compared to earlier approaches such as GANs, Normalizing Flows (NFs) (Dinh et al., 2016) and Variational Autoencoders (VAEs) (Kingma & Welling, 2013). Numerous researchers have applied diffusion and flow-based models to Real-ISR (Sahak et al., 2023; Yue et al., 2024). A notable direction further adapts large-scale text-to-image (T2I) diffusion models (Podell et al., 2023; Esser et al., 2024). These methods (Lin et al., 2023; Yu et al., 2024; Wu et al., 2024b; Duan et al., 2025) have achieved superior performance. However, due to the iterative denoising mechanism of diffusion and flow-based models, the inference process is computationally expensive. Thus, reducing the number of inference steps while maintaining sample quality has become a key challenge.

To address this, various one-step distillation methods have been proposed (Wang et al., 2024b; Wu et al., 2024a; Dong et al., 2025; You et al., 2025). Broadly, these methods either (i) match the output distribution between student and teacher models (Wu et al., 2024a; Dong et al., 2025), or (ii) constrain the student's denoising trajectory to remain consistent with that of the teacher (You et al., 2025). Although effective to some extent, existing methods often fail to recover fine details and completely lose the flexibility of few-step sampling.

More recently, MeanFlow (Geng et al., 2025) has emerged as an effective generative modeling paradigm. Unlike traditional flow models, which regress instantaneous velocity at each time step, MeanFlow instead targets the average velocity. It establishes an analytic relation, termed the Mean-Flow Identity that links average and instantaneous velocities via a time derivative. This formulation provides a principled training objective, avoiding heuristic consistency constraints and offering clear physical interpretation. At inference, MeanFlow supports flexible sampling strategies, allowing the model to map noisy state to any future point along the PF-ODE in a single step. Such flexibility is largely absent in existing one-step image restoration methods, making MeanFlow a natural foundation for developing a more versatile and tunable Real-ISR framework.

Although MeanFlow was originally proposed as a generative model trained from scratch, we argue that a two-stage strategy—first pre-training a teacher and then distilling into a student—is more effective and efficient. Directly learning both instantaneous and average velocities often leads to slow convergence, as the network struggles to learn shortcuts based on instantaneous velocity which it has not yet accurately captured. By contrast, distillation from a pre-trained teacher instead leverages already well-learned, high-quality instantaneous velocity field, thereby enabling faster convergence. This perspective is consistent with recent studies that emphasize the advantages of two-stage distillation (Lu & Song, 2024; Geng et al., 2024; Peng et al., 2025).

In this paper, we therefore treat MeanFlow as a distillation strategy to accelerate a powerful multi-step model into a one-step student network. To enhance performance, we propose a novel Classifier-Free Guidance (CFG)-based distillation strategy (Ho & Salimans, 2022): the teacher's CFG-enhanced prediction is used as the instantaneous velocity in the MeanFlow distillation loss. This modification yields stronger guidance and better performance than the original MeanFlow CFG formulation.

Unlike previous one-step SR approaches, our method, **Mean Flows for Super-Resolution (MFSR)**, does not rely on complex loss combinations to ensure restoration quality. It employs only the Mean-Flow distillation loss, computed entirely in the latent space. Consequently, gradients do not back-propagate through the encoder or decoder, unlike in (Wu et al., 2024a; Dong et al., 2025; Zhang et al., 2024), which significantly improves training efficiency. MFSR not only delivers high-quality one-step restoration, but also preserves the flexibility of few-step sampling, enabling a controllable trade-off between inference efficiency and restoration quality.

As shown in the left panel of Fig. 1, MFSR is capable of producing visually pleasing restorations with both high fidelity and perceptual realism in a single forward pass. Experiments on synthetic and real-world benchmarks demonstrate that our approach achieves superior restoration quality while being significantly faster than the teacher model. Our contributions are summarized as follows:

- We propose MFSR, the first framework that adapts MeanFlow to Real-ISR, enabling both one-step and few-step image restoration.
- We introduce a CFG-based MeanFlow distillation strategy that leverages the teacher's prior, yielding stronger supervision and better results than the original MeanFlow CFG formulation.
- Extensive experiments on synthetic and real-world benchmarks demonstrate that MFSR delivers strong perceptual quality, robust generalization, and efficient inference.

## 2 RELATED WORKS

### 2.1 FEW-STEP DIFFUSION/FLOW MODELS

Despite their strong generative power, diffusion models suffer from high inference cost. This motivates research on reducing sampling steps. For acceleration, existing distillation methods can be broadly categorized into two paradigms: distribution-based (Wang et al., 2024c; Yin et al., 2024b;a; Xu et al., 2024; Zhou et al., 2024b;a; Nguyen & Tran, 2024) and trajectory-based (Luhman & Luhman, 2021; Song et al., 2023; Salimans & Ho, 2022; Kim et al., 2023; Frans et al., 2024; Lu & Song, 2024). Distribution-based approaches (e.g., score distribution matching) (Wang et al., 2024c; Yin et al., 2024b)) aim to align the output distributions of student and teacher models. However, they often suffer from high computational cost, as they rely on an fake score model and alternate optimization between the student and the fake score network. Trajectory-based methods train the student with regression objectives derived from the PF-ODE. A representative method, Consistency Model (Song et al., 2023; Lu & Song, 2024), employs a loss function that constrains student predictions on two consecutive points along the same PF-ODE, ensuring coherent output across different timesteps. MeanFlow also belongs to the trajectory-based category, and we defer a detailed discussion to §3.2.

### 2.2 DIFFUSION/FLOW-BASED REAL-ISR

**Multi-step Diffusion-based Real-ISR.** Diffusion models have achieved remarkable success in the field of image super-resolution. Recent advances leverage powerful pre-trained text-to-image (T2I) models such as Stable Diffusion (SD) (Rombach et al., 2022) to address the challenges of Real-ISR (Wang et al., 2024a; Wu et al., 2024b; Yang et al., 2023; Yu et al., 2024; Duan et al., 2025). These methods typically guide or control the diffusion process to generate images that preserve the semantic content of degraded inputs while removing degradations. Representative works include SUPIR (Yu et al., 2024), which demonstrates strong generative ability by incorporating negative prompts and scaling up pre-training with larger models and datasets. Nevertheless, all of these methods remain limited by the multi-step denoising process inherent to diffusion models, which typically requires 20-50 denoising steps at inference. Besides, the employment of CFG needs 2 Number of Function Evaluations (NFEs) at each step, doubling the inference time.

**One-step Diffusion-based Real-ISR.** To reduce inference cost, several works have explored distillation techniques for Real-ISR. SinSR (Wang et al., 2024b) reformulates the inference process of ResShift (Yue et al., 2024) as an ODE and performs consistency-preserving distillation. CTMSR (You et al., 2025) applies Consistency Training (CT) (Song et al., 2023) and Distribution Trajectory Matching (DTM) to map perturbed LR inputs to HR in a single step. Yet these approaches remain constrained by the lack of large-scale training data. Another line of research focuses on score distillation. OSEDiff (Wu et al., 2024a) introduces the Variational Score Distillation (VSD) (Wang et al., 2024c) loss to Real-ISR tasks, achieving decent one-step performance by leveraging prior knowledge from pre-trained models. TSD-SR (Dong et al., 2025) further proposes Target Score Distillation (TSD), effectively addressing the issue of unreliable gradient direction caused by VSD. However, they both need to load an auxiliary score model and alternately train the student and score network, which increases the training overhead.

## 3 PRELIMINARY

### 3.1 RECTIFIED FLOW

Rectified Flow (Liu et al., 2022; Liu, 2022; Lipman et al., 2022; Albergo et al., 2023) is an ODE-based generative modeling framework. Given an initial distribution $\pi_0$ and a target data distribution

$\pi_1$, it learns a neural velocity field $v$ by minimizing:

$$\mathcal{L}_{\text{RF}} = \mathbb{E}_{x_0 \sim \pi_0, x_1 \sim \pi_1} \left[ \int_0^1 \left\| v(x_t, t) - (x_1 - x_0) \right\|^2 \mathrm{d}t \right], \quad \text{with} \quad x_t = (1 - t)x_0 + tx_1, \quad (1)$$

where $x_t$ is the linear interpolation of $x_0$ and $x_1$. After training, sample generation reduces to solving the following neural ODE:

$$\frac{\mathrm{d}x_t}{\mathrm{d}t} = v(x_t, t), \quad t \in [0, 1], \tag{2}$$

which can be numerically approximated using standard ODE solvers. For instance, applying the first-order Euler method yields:

$$x_{t + \frac{1}{N}} = x_t + \frac{1}{N} v(x_t, t), \quad t \in \{0, 1, \dots, N - 1\}/N. \tag{3}$$

Here, the trajectory is integrated in $N$ steps with a step size of $1/N$. A larger $N$ provides higher accuracy at the expense of slower sampling, while a smaller $N$ accelerates generation but reduces sample quality.

## 3.2 MEANFLOW

Unlike standard Rectified Flow, which learns an instantaneous velocity field, MeanFlow (Geng et al., 2025) regresses the average velocity field over an interval. Specifically, given a time interval $[t, s]$, the model will take a current state $x_t$ as input and defines a vector pointing to the next state $x_s$ ($s > t$) via:

$$x_s = x_t + (s - t)u(x_t, t, s), \tag{4}$$

where $u$ is the average velocity, defined by $u(x_t, t, s) = \frac{1}{s-t} \int_t^s v(x_\tau, \tau) \mathrm{d}\tau$. By differentiating both sides on Eq. (4) with respect to $t$ and re-arranging terms, one can obtain the *MeanFlow Identity*, which describes the relation between average velocity $u(x_t, t, s)$ and instantaneous velocity $\frac{\mathrm{d}x_t}{\mathrm{d}t}$:

$$u(x_t, t, s) = \frac{\mathrm{d}x_t}{\mathrm{d}t} + (s - t)\frac{\mathrm{d}u(x_t, t, s)}{\mathrm{d}t}. \tag{5}$$

The derivative $\frac{\mathrm{d}u(x_t,t,s)}{\mathrm{d}t}$ can be expanded by its partial components, $\frac{\mathrm{d}u(x_t,t,s)}{\mathrm{d}t} = \frac{\partial u(x_t,t,s)}{\partial x_t}\frac{\mathrm{d}x_t}{\mathrm{d}t} + \frac{\partial u(x_t,t,s)}{\partial t}$, which corresponds to a Jacobian-Vector Product (JVP). Then we minimize this objective:

$$\mathcal{L}_{\text{MF}} = \mathbb{E}_{x_0, x_1, t, s} \left\| u(x_t, t, s) - \text{sg}(u_{\text{tgt}}) \right\|_2^2, \quad \text{with} \quad u_{\text{tgt}} = \frac{\mathrm{d}x_t}{\mathrm{d}t} + (s - t)\left[ \frac{\partial u(x_t, t, s)}{\partial x_t}\frac{\mathrm{d}x_t}{\mathrm{d}t} + \frac{\partial u(x_t, t, s)}{\partial t} \right], \quad (6)$$

where $u_{\text{tgt}}$ serves as the *effective regression target*, $\text{sg}(\cdot)$ denotes stop-gradient operation, and the JVP term can be calculated approximately at the same cost of one forward operation. During sampling, the numerical integration of instantaneous velocity $\int_t^s v(x_\tau, \tau)\mathrm{d}\tau$ in Rectified Flow can be replaced by $(s - t)u(x_t, t, s)$. In the case of 1-step sampling, one can simply have $x_1 = x_0 + u(x_0, 0, 1)$, where $x_0$ is sampled from an initial distribution $\pi_0$.

## 3.3 DIT4SR

DiT4SR (Duan et al., 2025) builds on Stable Diffusion3.5 (SD3.5) (Esser et al., 2024), a large-scale Rectified Flow model that employs Diffusion Transformers (DiTs) (Peebles & Xie, 2023) as backbone. To adapt SD3.5 for Real-ISR, DiT4SR integrates a LR stream into the DiT blocks, enabling high perceptual realism in the restored images. During inference, DiT4SR starts from Gaussian noise and performs iterative denoising conditioned on the latent LR image and a text prompt extracted from it. Formally, the DiT4SR sampling process is described by the PF-ODE:

$$\frac{\mathrm{d}z_t}{\mathrm{d}t} = v(z_t, t \mid z_{\text{LR}}, c), \tag{7}$$

where $z_t = tz_{\text{HR}} + (1 - t)\epsilon$, $z_{\text{HR}}$ is the latent HR image, $\epsilon$ is Gaussian noise, and $c$ denotes the text prompt. DiT4SR typically requires about 40 denoising steps to produce high-quality reconstructions, and reducing the number of steps leads to a significant drop in performance.

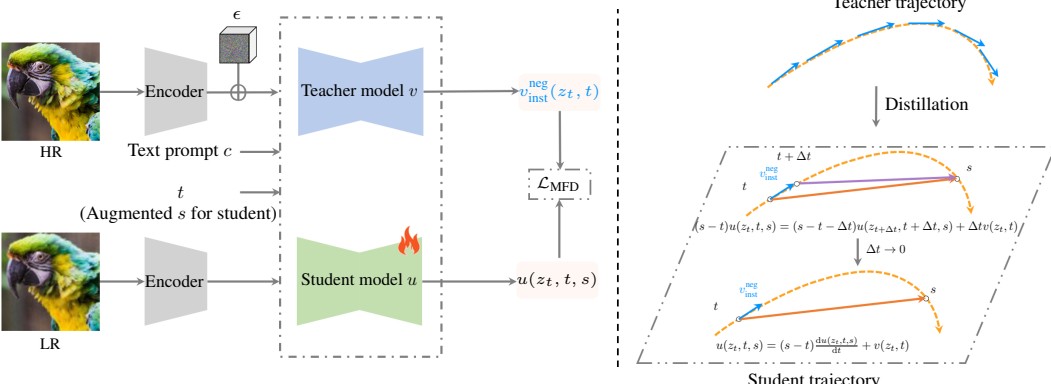

Figure 2: Overview of the MeanFlow Distillation (MFD) pipeline. **Left:** Training pipeline where the student model is initialized from the teacher. Student model need two time steps as input. **Right:** Derivation of the MFD loss. The dotted **orange** line denotes the teacher's and student's PF-ODE, the **blue** line shows the teacher's predicted instantaneous velocity, the solid **orange** line shows the student's predicted average velocity and the **purple** line indicates the student's predicted average velocity over a shorter interval. Taking the limit $\Delta t \to 0$ yields the MFD loss $\mathcal{L}_{\text{MFD}}$.

## 4 METHOD

### 4.1 FRAMEWORK OVERVIEW

Our goal is to distill a powerful but slow multi-step teacher into a one/few-step MeanFlow student for Real-ISR. We adopt DiT4SR (Duan et al., 2025) as the teacher model. Overview of the distillation pipeline is shown in Fig. 2. The framework comprises four components: a visual encoder $E$, a prompt extractor, a teacher model $v$, and a student model $u$, with only $u$ being trainable. Given a high-low resolution image pair $(x_{\text{HR}}, x_{\text{LR}})$, we first sample time steps $t, s$ and random noise $\epsilon$. A text prompt $c$ is extracted from $x_{\text{HR}}$, and both $x_{\text{HR}}$ and $x_{\text{LR}}$ are encoded into latent representations $z_{\text{HR}}$ and $z_{\text{LR}}$. We then interpolate $z_{\text{HR}}$ with $\epsilon$ according to $t$ to obtain $z_t$, with boundary states $z_0 = \epsilon$ and $z_1 = z_{\text{HR}}$. Conditioned on $z_{\text{LR}}$ and prompt $c$, the teacher receives $(z_t, t)$ while the student receives $(z_t, t, s)$ to compute the MeanFlow Distillation loss $\mathcal{L}_{\text{MFD}}$. The loss is calculated in latent space, therefore no decoding is needed during training. In the following sections, we describe the detailed designs of MFSR and its training loss.

### 4.2 MODEL INITIALIZATION AND TIMESTEP AUGMENTATION

While MeanFlow can be trained from scratch, initializing from a pre-trained DiT4SR teacher is far more efficient and practical. Since the teacher produces high-quality restorations, it offers a reliable trajectory from which the student can learn effective shortcuts.

The teacher predicts the *instantaneous velocity* $\frac{\mathrm{d}z_t}{\mathrm{d}t}$ at a single time step, requiring only one time embedding. In contrast, our student is designed to predict the *average velocity* over an interval $[t, s]$, which requires the start and end timestep of the interval to avoid ambiguity.

To accommodate this, we augment the original DiT4SR architecture with an additional time-embedding branch so that the student can also take the end time $s$ as input. Specifically, we duplicate the network structure of the original $t$-time embedder from DiT4SR and use this copy as a separate $s$-time embedder. The resulting $s$-embedding is added to the $t$-embedding before being fed into the network. The teacher $v(z_t, t)$ is thus adapted into a student model $u(z_t, t, s)$. Then, extending the unconditional *MeanFlow Identity* (Eq. (5)) to the LR- and text-conditioned case yields:

$$u(z_t, t, s \mid z_{\text{LR}}, c) = \frac{\mathrm{d}z_t}{\mathrm{d}t} + (s - t)\left[\frac{\partial u(z_t, t, s \mid z_{\text{LR}}, c)}{\partial z_t}\frac{\mathrm{d}z_t}{\mathrm{d}t} + \frac{\partial u(z_t, t, s \mid z_{\text{LR}}, c)}{\partial t}\right] \quad (8)$$

### 4.3 IMPROVED INSTANTANEOUS VELOCITY WITH TEACHER CFG

A crucial component in MeanFlow distillation is the choice of the instantaneous velocity $\frac{\mathrm{d}z_t}{\mathrm{d}t}$. A naïve choice is to use the Ground-Truth (GT) velocity $z_1 - z_0$. Under this circumstance, the teacher model is just used as initialization for student model. However, we empirically observe that this

leads to inferior restoration results. Notably, the teacher achieves strong visual realism by employing CFG, suggesting that CFG plays a crucial role in semantic alignment and perceptual quality.

The original MeanFlow paper attempts to enhance the velocity field using the student model itself under CFG. In the Real-ISR setting, it can be formulated as:

$$v_{\text{inst}}^{\text{orig}} = w(z_1 - z_0) + \kappa u(z_t, t, t \mid z_{\text{LR}}, c) + (1 - w - \kappa)u(z_t, t, t \mid z_{\text{LR}}, c = \varnothing), \quad (9)$$

where $w$ and $\kappa$ are scalar weighting factor, and the effective guidance scale is $w' = \frac{w}{1-\kappa}$. However, since the student is still being optimized, this self-referential target hampers convergence.

In our setting, however, a pre-trained teacher model is available. This provides a better alternative: instead of relying on the student's self-improvement, we directly use the teacher's CFG-based prediction to construct the instantaneous velocity. Concretely, we define:

$$v_{\text{inst}}^{\text{null}} = v(z_t, t \mid z_{\text{LR}}, c) + w\big(v(z_t, t \mid z_{\text{LR}}, c) - v(z_t, t \mid z_{\text{LR}}, c = \varnothing)\big). \quad (10)$$

This formulation incorporates the semantic prior from the text prompt through the teacher's guidance, while maintaining stability during training.

Furthermore, we extend this idea by incorporating negative prompts into the teacher's CFG. Prior works (Yu et al., 2024; Zhang et al., 2024) have shown that negative prompts [1] can effectively suppress undesired artifacts and improve the perceptual quality of generated images. Inspired by this, we replace the null condition in Eq. (10) with a negative prompt condition, leading to:

$$v_{\text{inst}}^{\text{neg}} = v(z_t, t \mid z_{\text{LR}}, c) + w\big(v(z_t, t \mid z_{\text{LR}}, c) - v(z_t, t \mid z_{\text{LR}}, c^{\text{neg}})\big). \quad (11)$$

Compared with Eq. (10), this formulation provides stronger supervision by explicitly discouraging unrealistic or low-quality attributes, thereby encouraging the student model to generate sharper details and richer textures.

Finally, the MeanFlow Distillation loss integrates teacher-guided instantaneous velocity:

$$\mathcal{L}_{\text{MFD}}(\theta) = \mathbb{E}_{z_{\text{HR}}, z_{\text{LR}}, \epsilon, t, s}\big\|u(z_t, t, s \mid z_{\text{LR}}, c) - \text{sg}(u_{tgt})\big\|_2^2,$$
$$\text{with } u_{tgt} = v_{\text{inst}}^{\text{neg}} + (s - t)\Big[\frac{\partial u_\theta(z_t, t, s \mid z_{\text{LR}}, c)}{\partial z_t} v_{\text{inst}}^{\text{neg}} + \frac{\partial u_\theta(z_t, t, s \mid z_{\text{LR}}, c)}{\partial t}\Big]. \quad (12)$$

During inference, the student model takes the LR image and the extracted text prompt as conditioning inputs. We perform $N$-step sampling with uniformly spaced timesteps $0 = \tau_1 < \tau_2 < \cdots < \tau_N = 1$, starting from initial noise $z_0$. The update at each step is given by $z_{\tau_{n+1}} = z_{\tau_n} + (\tau_{n+1} - \tau_n)u\big(z_{\tau_n}, \tau_n, \tau_{n+1} \mid z_{\text{LR}}, c\big)$.

## 4.4 DESIGN DECISIONS

**Stabilizing Time Embedding.** In the case of distilling DiT4SR, naively computing the Jacobian-Vector Product (JVP) term $\frac{\mathrm{d}u(z_t, t, s)}{\mathrm{d}t}$ often leads to training instabilities. As shown in (Lu & Song, 2024; Chen et al., 2025), the time-derivative can be decomposed as $\partial_t u = \frac{\partial c_{\text{noise}}(t)}{\partial t} \cdot \frac{\partial \text{emb}(c_{\text{noise}})}{\partial c_{\text{noise}}} \cdot \frac{\partial u}{\partial \text{emb}(c_{\text{noise}})}$, where $\text{emb}(\cdot)$ denotes the time embeddings and $c_{noise}(\cdot)$ is time transformation. In prior Rectified Flow models such as SD3.5, the choice $c_{\text{noise}}(t) = 1000t$ amplifies the time derivative $\partial_t u$ by a factor of 1000, resulting in large fluctuations during training. To mitigate this issue, we adopt the remedy proposed in (Lu & Song, 2024; Chen et al., 2025) and set $c_{\text{noise}}(t) = t$ in the student model. This modification avoids excessive amplification of gradient norms and yields more stable training dynamics. Note that the teacher model does not need this modification, as the time-derivative computation does not propagate through its architecture.

**Sampling Time Steps.** We draw two time steps $(t, s)$ from the joint distribution $p(t, s) = p(t)p(s \mid t)$, where $p(t) = \mathcal{U}[0, 1]$ and $p(s \mid t) = \mathcal{U}[t, 1]$. Following (Geng et al., 2025), we enforce a certain portion of $t = s$. Specifically, when $t = s$, the model learns the instantaneous velocity, while when $t \neq s$, it learns the shortcut between time steps (average velocity).

**Loss Metrics.** Instead of the squared L2 loss or adaptive L2 loss used in (Geng et al., 2025), we use Pseudo-Huber loss as suggested in (Song & Dhariwal, 2023) to reduce loss variance during training.

---

[1]"oil painting, cartoon, blur, dirty, messy, low quality, deformation, low resolution, oversmooth."

Table 1: Quantitative comparison with the state-of-the-art one-step methods across both synthetic and real-world benchmarks. The number of diffusion inference steps is indicated by 's'. The best and second best results of each metric are highlighted in red and blue, respectively.

| Datasets | Method | PSNR↑ | SSIM↑ | LPIPS↓ | DIST↓ | FID↓ | NIQE↓ | MUSIQ↑ | MANIQA↑ | CLIPIQA↑ |
|---|---|---|---|---|---|---|---|---|---|---|
| DRealSR | OSEDiff-1s | 27.92 | 0.7836 | 0.2966 | 0.2163 | 135.39 | 6.4381 | 64.67 | 0.5898 | 0.6959 |
| | AddSR-1s | 27.77 | 0.7722 | 0.3196 | 0.2242 | 150.18 | 6.9321 | 60.85 | 0.5490 | 0.6188 |
| | SinSR-1s | 28.27 | 0.7465 | 0.3730 | 0.2501 | 182.28 | 7.0246 | 55.55 | 0.4907 | 0.6391 |
| | CTMSR-1s | 28.66 | 0.7838 | 0.3232 | 0.2357 | 162.29 | 6.1426 | 59.84 | 0.4865 | 0.6505 |
| | S3Diff-1s | 27.53 | 0.7491 | 0.3109 | 0.2100 | 118.49 | 6.2142 | 63.94 | 0.6124 | 0.7132 |
| | TSDSR-1s | 26.19 | 0.7170 | 0.3116 | 0.2204 | 130.70 | 5.7643 | 66.11 | 0.5820 | 0.7303 |
| | MFSR-1s | 24.15 | 0.6423 | 0.3660 | 0.2379 | 143.12 | 6.0241 | 64.47 | 0.6148 | 0.7171 |
| | MFSR-2s | 24.29 | 0.6455 | 0.3689 | 0.2333 | 139.56 | 6.2711 | 64.45 | 0.6354 | 0.7023 |
| RealSR | OSEDiff-1s | 25.15 | 0.7341 | 0.2920 | 0.2128 | 123.57 | 5.6345 | 69.09 | 0.6335 | 0.6685 |
| | AddSR-1s | 24.79 | 0.7077 | 0.3091 | 0.2191 | 132.05 | 5.5440 | 66.18 | 0.6098 | 0.5722 |
| | SinSR-1s | 26.23 | 0.7342 | 0.3191 | 0.2363 | 136.65 | 6.2773 | 60.84 | 0.5418 | 0.6224 |
| | CTMSR-1s | 25.98 | 0.7543 | 0.2901 | 0.2209 | 135.69 | 5.5046 | 64.49 | 0.5276 | 0.6397 |
| | S3Diff-1s | 25.18 | 0.7269 | 0.2721 | 0.2005 | 105.12 | 5.2708 | 67.82 | 0.6424 | 0.6734 |
| | TSDSR-1s | 23.40 | 0.6886 | 0.2805 | 0.2183 | 114.56 | 5.0924 | 70.76 | 0.6312 | 0.7198 |
| | MFSR-1s | 21.51 | 0.6347 | 0.3158 | 0.2295 | 110.14 | 5.2421 | 67.95 | 0.6389 | 0.6968 |
| | MFSR-2s | 21.75 | 0.6494 | 0.2999 | 0.2222 | 107.87 | 5.5980 | 67.45 | 0.6560 | 0.6705 |
| DIV2K-Val | OSEDiff-1s | 23.86 | 0.6233 | 0.2896 | 0.1999 | 100.53 | 4.9741 | 68.53 | 0.6111 | 0.6692 |
| | AddSR-1s | 22.39 | 0.5652 | 0.3728 | 0.2387 | 133.78 | 5.9929 | 63.39 | 0.5657 | 0.5734 |
| | SinSR-1s | 24.50 | 0.6136 | 0.3164 | 0.2110 | 131.96 | 6.1721 | 64.26 | 0.5442 | 0.6687 |
| | CTMSR-1s | 24.87 | 0.6349 | 0.3011 | 0.2102 | 126.49 | 5.3036 | 66.59 | 0.5146 | 0.6602 |
| | S3Diff-1s | 23.68 | 0.6075 | 0.2545 | 0.1759 | 84.92 | 5.0358 | 68.40 | 0.6252 | 0.7012 |
| | TSDSR-1s | 22.17 | 0.5680 | 0.2679 | 0.1901 | 103.49 | 4.6621 | 71.19 | 0.6010 | 0.7221 |
| | MFSR-1s | 21.25 | 0.5479 | 0.3143 | 0.2029 | 111.45 | 4.5831 | 69.30 | 0.6256 | 0.7199 |
| | MFSR-2s | 21.49 | 0.5626 | 0.2965 | 0.1933 | 106.09 | 4.8895 | 68.34 | 0.6364 | 0.6906 |

## 5 EXPERIMENTS

### 5.1 EXPERIMENTAL SETTINGS

**Training Datasets.** We construct the training set using a combination of images from DIV2K (Agustsson & Timofte, 2017), DIV8K (Gu et al., 2019), Flickr2K (Timofte et al., 2017), LSDIR (Li et al., 2023), NKUSR8K (Duan et al., 2025), and the first 10K face images from FFHQ (Karras et al., 2019). To generate paired data, we apply the Real-ESRGAN (Wang et al., 2021) degradation pipeline. The resolution of resulting LR and HR images are set to $128 \times 128$ and $512 \times 512$, respectively.

**Test Datasets.** We evaluate performance on both synthetic and real-world datasets. The synthetic set contains 100 randomly cropped $512 \times 512$ images from the DIV2K validation set and degrade using the Real-ESRGAN pipeline. For real-world evaluation, we employ RealSR (Cai et al., 2019), DRealSR (Wei et al., 2020), RealLR200 (Wu et al., 2024b), and RealLQ250 (Ai et al., 2025) datasets. All experiments are conducted with the scaling factor of $\times 4$. Center-cropping is applied to RealSR and DRealSR, and the resolution of their LR images is set to $128 \times 128$. Both RealLR200 and RealLQ250 lack corresponding GT images, and no cropping is performed on these two datasets.

**Evaluation Metrics.** To evaluate our method, we adopt both reference-based and no-reference metrics. Reconstruction fidelity is measured using PSNR and SSIM (Wang et al., 2004), while perceptual similarity is assessed with LPIPS (Zhang et al., 2018) and DISTS (Ding et al., 2020). In addition, FID (Heusel et al., 2017) is used to quantify the distributional discrepancy between restored and GT images. For no-reference Image Quality Assessment (IQA), we include NIQE (Zhang et al., 2015), CLIPIQA (Wang et al., 2023), MUSIQ (Ke et al., 2021), and MANIQA (Yang et al., 2022); for datasets lacking ground truth, we additionally employ LIQE (Zhang et al., 2023). It is worth noting that quantitative metrics only partially capture perceptual quality, as prior studies have shown that these metrics often diverge from human judgments (Jinjin et al., 2020; Yu et al., 2024; Lin et al., 2025). Therefore, we report these metrics just for reference and mainly focus on user study.

**Compared Methods.** We compare our method with several one-step diffusion-based methods SinSR (Wang et al., 2024b), CTMSR (You et al., 2025), OSEDiff (Wu et al., 2024a), AddSR (Xie et al., 2024), S3Diff (Zhang et al., 2024), TSDSR (Dong et al., 2025). Comparison with multi-step diffusion-based methods can be found in the **Supplementary Material**.

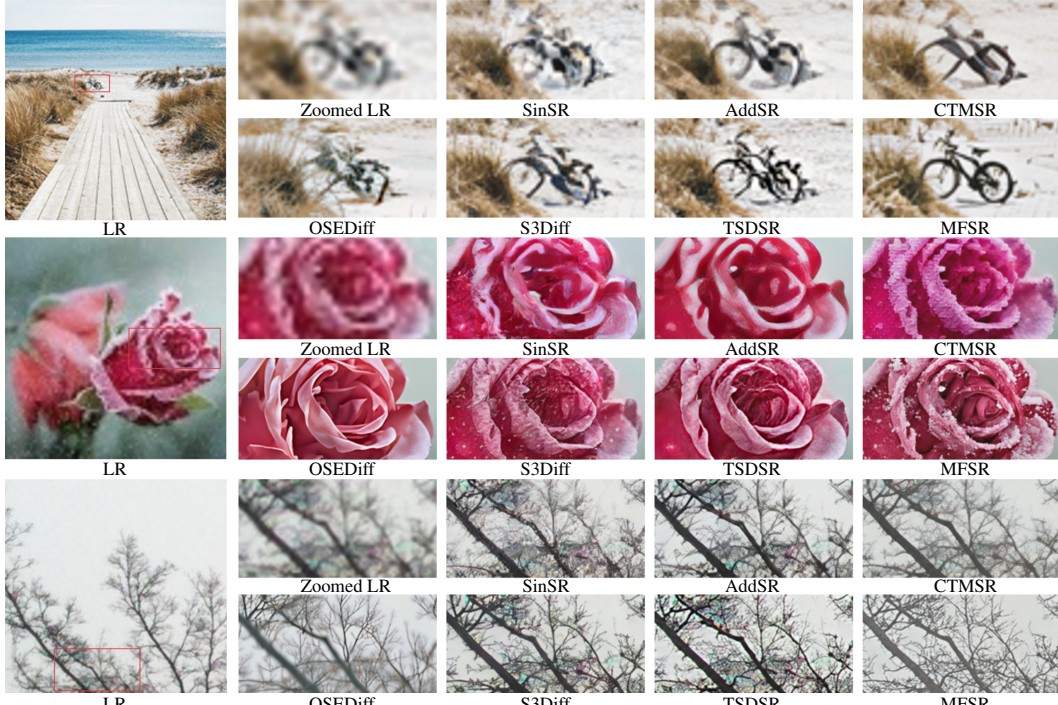

Figure 3: Qualitative comparison with state-of-the-art methods. All methods perform 1-step inference. Our MFSR is capable of generating vivid details without artifacts or remaining degradations.

## 5.2 COMPARISON WITH EXISTING METHODS

**Qualitative Comparisons.** Fig. 3 presents visual comparisons with other one-step baselines. In the first row, our method demonstrates a clear advantage in recovering fine structural details of the bicycle. In the second row, it successfully generates rich textures (e.g., frost and snow covering the flower), benefiting from the strong generative prior initialized from the teacher model. While OSEDiff produces artifact-free outputs, its results are noticeably over-smoothed. In the last row, our method effectively removes undesired degradation patterns, whereas competing approaches still suffer from blurring and color distortions. These results highlight the superiority of our MeanFlow distillation framework in achieving both structural fidelity and perceptual realism.

**Quantitative Comparisons.** Tables 1 and 2 report the quantitative results. The relatively lower PSNR/SSIM scores can be attributed to the perception-distortion (realism-fidelity) trade-off (Blau & Michaeli, 2018; Zhu et al., 2024). Notably, our method achieves leading MANIQA score with one-step sampling, and further improves with two-steps. It also shows competitive performance on FID, NIQE, MUSIQ, and CLIPIQA, though not always the best. Since quantitative metrics are often misaligned with human perception in generative restoration, we present them mainly for reference and place greater emphasis on the user study, which more faithfully reflects perceptual quality.

## 5.3 USER STUDY

To further assess perceptual quality, we conduct a user study with 75 volunteers. We randomly sampled 25 LR images from RealLQ250, and compared 1-step MFSR against four representative methods: SinSR, CTMSR, OSEDiff, and TSDSR. For each image, participants were asked to select the restoration that best balances realism of textures and details and structural fidelity to the LR input. The percentage of votes (preference rate) obtained by each method is reported in Fig. 4. MFSR received the highest preference rate of 38.9%, significantly outperforming the second-best method. These results confirm that MFSR delivers the most perceptually preferred results.

Table 2: Quantitative comparison with the state-of-the-art one-step methods on real-world benchmarks lacking ground-truth image. The number of diffusion inference steps is indicated by 's'.

Figure 4: Results of user study, with numbers showing vote percentages for each method.

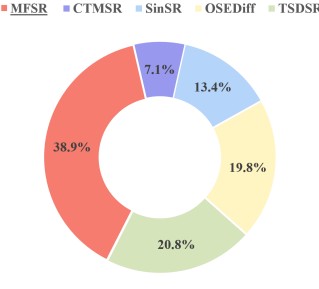

| Datasets | Method | NIQE ↓ | MUSIQ ↑ | MANIQA ↑ | CLIPIQA ↑ | LIQE ↑ |
|---|---|---|---|---|---|---|
| RealLQ250 | OSEDiff-1s | 3.9656 | 69.55 | 0.5782 | 0.6725 | 3.9039 |
| | SinSR-1s | 5.8204 | 63.73 | 0.5161 | 0.6990 | 3.2578 |
| | CTMSR-1s | 4.5835 | 68.00 | 0.5078 | 0.6706 | 3.3373 |
| | S3Diff-1s | 3.9715 | 69.19 | 0.6016 | 0.7043 | 4.0192 |
| | TSDSR-1s | **3.4868** | **72.09** | 0.5829 | **0.7221** | 4.0834 |
| | MFSR-1s | 3.5309 | 70.65 | 0.6040 | 0.6992 | **4.2136** |
| | MFSR-2s | 3.5560 | 70.58 | **0.6204** | 0.7047 | 4.1687 |
| RealLR200 | OSEDiff-1s | 4.0199 | **69.60** | 0.6020 | 0.6752 | 4.0560 |
| | SinSR-1s | 5.5887 | 63.59 | 0.5421 | 0.6955 | 3.4758 |
| | CTMSR-1s | 4.2815 | 67.60 | 0.5354 | 0.6738 | 3.6061 |
| | S3Diff-1s | 4.0360 | 68.92 | 0.6172 | 0.7025 | 4.0643 |
| | TSDSR-1s | **3.6400** | **71.02** | 0.6093 | **0.7212** | 4.1035 |
| | MFSR-1s | 3.6690 | 69.50 | 0.6190 | 0.6893 | **4.1813** |
| | MFSR-2s | 3.7721 | 69.38 | **0.6344** | 0.6876 | 4.1564 |

Table 3: Ablation studies for CFG strategy and its scale $w$.

| Instantaneous Velocity | $w$ | LPIPS ↓ | DISTS ↓ | FID ↓ | NIQE ↓ | MUSIQ ↑ | MANIQA ↑ | CLIPIQA ↑ |
|---|---|---|---|---|---|---|---|---|
| $z_1 - z_0$ | - | 0.3210 | 0.2276 | 121.78 | 5.5784 | 65.32 | 0.6035 | 0.6474 |
| Original MeanFlow CFG | 6 | 0.3478 | 0.2453 | 120.81 | 5.8691 | 67.33 | 0.6091 | **0.6951** |
| Ours null | 6 | **0.2931** | **0.2151** | **109.75** | 5.7146 | 65.27 | 0.6184 | 0.6551 |
| Ours neg | 1 | **0.2983** | **0.2237** | **109.23** | **5.2234** | 66.84 | 0.6273 | 0.6747 |
| Ours neg | 4 | 0.3021 | 0.2255 | 110.83 | 5.2317 | 67.02 | 0.6317 | 0.6773 |
| Ours neg | 6 | 0.3158 | 0.2295 | 110.14 | 5.2421 | **67.95** | **0.6389** | **0.6968** |
| Ours neg | 8 | 0.3151 | 0.2300 | 110.01 | **5.2253** | **67.69** | **0.6364** | 0.6879 |

## 5.4 EFFECT OF INCREASING INFERENCE STEPS

Our one-step results already surpass existing one-step baselines. Furthermore, unlike prior methods, our framework supports few-step inference. In Fig. 5, we evaluate the effect of different sampling steps on RealLQ250 and report MANIQA. S3Diff and TSDSR perform 1-step sampling. Increasing the step count from one to two brings a clear improvement, demonstrating the benefit of optional refinement. Extending the steps to three, four or five yields moderate improvements, while increasing to eight steps results in only marginal gains. These results show that most perceptual benefits are captured within the first few steps.

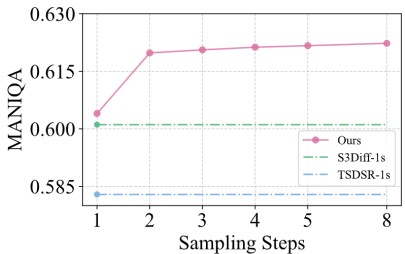

Figure 5: Effect of sampling steps.

## 5.5 ABLATION STUDY

**Effectiveness of CFG strategy.** We evaluate our proposed CFG strategy for MeanFlow distillation by comparing different instantaneous velocity formulations on RealLQ250, including the GT field ($z_1 - z_0$), the original MeanFlow CFG strategy, and our CFG variants with null and negative prompts. We also conduct an ablation study on the CFG scale $w$. The original MeanFlow CFG have an effective guidance scale of $w' = \frac{w}{1-\kappa} = 6$, with $w = 1$ and $\kappa = 0.83$ in Eq. (9). As shown in Tab. 3, our strategy achieves the best quality scores (MUSIQ, MANIQA, CLIPIQA), demonstrating its effectiveness over baselines. Among different CFG scales, $w = 6$ with negative prompt yields the best performance, and is therefore adopted as our default configuration. Additional visual comparisons are provided in the **Supplementary Material**.

## 6 CONCLUSION

In this paper, we propose Mean Flows for Super-Resolution (MFSR), a effective distillation method that enables high-realism restoration results in only one step while retaining the option of few-step sampling to trade compute for sample quality. We adapt MeanFlow to distill a multi-step Real-ISR teacher into student model. To improve SR performance, we make modifications to original MeanFlow CFG strategy to achieve stronger guidance and better performance. Extensive experiment results demonstrate the effectiveness of our method, highlighting its ability to restore fine details with remarkable realism.

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

## A ALGORITHM

The pseudo-code of MFSR training and inference algorithm is summarized as 1 and 2.

---

**Algorithm 1** MFSR training

---

**Require:** Pre-trained teacher model $v$, VAE encoder $E$, prompt extractor $Y$, data distribution $p_D$, two time step joint distribution $p_T$, stop gradient operator sg[·], a predefined metric function $d(\cdot, \cdot)$
1: Student model $u \leftarrow \text{copyWeights}(v)$,   // intialize
2: Add the second time embedder to $u$
3: **repeat**
4:     Sample $\epsilon \sim \mathcal{N}(0, 1)$, $(x_{\text{HR}}, x_{\text{LR}}) \sim p_D$, $t, s \sim p_T$
5:     Calculate $z_{\text{HR}}, z_{\text{LR}} = E(x_{\text{HR}}), E(x_{\text{LR}})$
6:     Calculate $c = Y(x_{\text{HR}})$
7:     Calculate $z_t = t z_{\text{HR}} + (1 - t)\epsilon$
8:     Calculate $\frac{dz_t}{dt} = v(z_t, t \mid z_{\text{LR}}, c) + w\big(v(z_t, t \mid z_{\text{LR}}, c) - v(z_t, t \mid z_{\text{LR}}, c^{\text{neg}})\big)$.
9:     Calculate loss $\mathcal{L} = d\left(u(z_t, t, s \mid z_{\text{LR}}, c), \text{sg}\left[\frac{dz_t}{dt} + (s - t)\frac{du(z_t, t, s \mid z_{\text{LR}}, c)}{dt}\right]\right)$
10:     Update $u$ with the loss gradient $\nabla \mathcal{L}$
11: **until** *convergence*
12: **Return** student model $u$

---

---

**Algorithm 2** MFSR inference

---

**Require:** MFSR model $u$, VAE encoder $E$, VAE decoder $D$, prompt extractor $Y$, LR image $x_{\text{LR}}$, sampling steps N, sequence of time points $0 = \tau_1 < \tau_2 < \cdots < \tau_N = 1$, initial noise $z_0$
1: Calculate $z_{\text{LR}} = E(x_{\text{LR}})$
2: Calculate $c = Y(x_{\text{LR}})$
3: **for** $n = 0$ to $N - 1$ **do**
4:     Calculate $z_{\tau_{n+1}} = z_{\tau_n} + (\tau_{n+1} - \tau_n)u(z_{\tau_n}, \tau_n, \tau_{n+1} \mid z_{\text{LR}}, c)$
5: **end for**
6: Calculate $\hat{x}_{\text{HR}} = D(z_1)$
7: **Return** super-resolved image $\hat{x}_{\text{HR}}$

---

## B IMPLEMENTATION DETAILS

Our model is initialized from the teacher model DiT4SR, which is built upon SD3.5. During training, we freeze the original parameters of SD3.5 and only update the additional parameters introduced by DiT4SR, as detailed in Sec.3 of the DiT4SR paper (Duan et al., 2025). Besides, we incorporate LoRA (Hu et al., 2021) into the transformer blocks of SD3.5, with a LoRA rank of 64. Following DiT4SR, we use LLaVA (Liu et al., 2024) as the prompt extractor. We employ the Adam optimizer with a learning rate of 5e-5. Training is conducted on 8 NVIDIA H200 GPUs with a batch size of 80, and the entire process takes approximately 19 hours. In total, the model is trained for 12K iterations.

## C DERIVATION OF THE CONTINUOUS-TIME EQUATION

Here we show detailed derivations of the continuous-time equations in Fig. 2. Starting from the original equation,

$$(s - t)\, u(z_t, t, s) = (s - t - \Delta t)\, u(z_{t+\Delta t},\, t + \Delta t,\, s) + \Delta t\, v(z_t, t). \tag{13}$$

Since $u$ is differentiable in both $z$ and $t$ and the path $t \mapsto z_t$ is differentiable, we can apply a first-order Taylor expansion of $u(z_{t+\Delta t}, t + \Delta t, s)$ along the trajectory $(z_t, t)$:

$$u(z_{t+\Delta t}, t + \Delta t, s) = u(z_t, t, s) + \Delta t\, \frac{d}{dt} u(z_t, t, s) + r(\Delta t), \tag{14}$$

where the remainder satisfies

$$\lim_{\Delta t \to 0} \frac{r(\Delta t)}{\Delta t} = 0, \qquad \text{i.e. } r(\Delta t) = o(\Delta t).$$

Substituting Eq. (14) into the right-hand side of Eq. (13) gives

$$(s - t)\, u(z_t, t, s) = (s - t - \Delta t)\left[u(z_t, t, s) + \Delta t\, \frac{d}{dt} u(z_t, t, s) + r(\Delta t)\right] + \Delta t\, v(z_t, t)$$

$$\cancel{(s-t)\, u(z_t, t, s)} = \cancel{(s-t)\, u(z_t, t, s)} - \Delta t\, u(z_t, t, s)$$
$$+ (s - t)\, \Delta t\, \frac{d}{dt} u(z_t, t, s) - \Delta t^2\, \frac{d}{dt} u(z_t, t, s)$$
$$+ (s - t) r(\Delta t) - \Delta t\, r(\Delta t) + \Delta t\, v(z_t, t).$$

Dividing through by $\Delta t$ (for $\Delta t \neq 0$) and re-arranging terms gives

$$u(z_t, t, s) = (s - t)\, \frac{d}{dt} u(z_t, t, s) - \Delta t\, \frac{d}{dt} u(z_t, t, s) + \frac{(s-t) r(\Delta t)}{\Delta t} - r(\Delta t) + v(z_t, t).$$

Taking the limit $\Delta t \to 0$, we have $\dfrac{r(\Delta t)}{\Delta t} \to 0$ and $r(\Delta t) \to 0$, while also noting $\Delta t\, \frac{d}{dt} u(z_t, t, s) \to 0$, we obtain the final result:

$$\boxed{u(z_t, t, s) = (s - t)\, \frac{d}{dt} u(z_t, t, s) + v(z_t, t)} \tag{15}$$

And this is used to construct MeanFlow Distillation loss in Eq. (12)

## D    NOISE VS. LR INITIALIZATION

We adopt Gaussian noise as the initial state for denoising. While recent works (Wu et al., 2024a; Dong et al., 2025) instead initialize from the LR image, we find that noise initialization offers clear advantages. First, it allows the model to synthesize richer details and textures, whereas starting from the LR image tends to restrict the generative capacity and makes it difficult to remove complex degradations (as shown in the third row of Fig. 3). Second, initializing from noise ensures consistency with the teacher's PF-ODE, thereby strengthening the student's ability to inherit the teacher's generative prior.

## E    DIFFERENCE FROM PREVIOUS WORKS

**Guided distillation (Meng et al., 2023).** Guided distillation, originally proposed for text-to-image generation, transfers knowledge from a teacher model with CFG to a few-step student model via a two-stage process. The first stage trains a model to match CFG-enhanced outputs of the teacher, and the second stage progressively distills it into a few-step diffusion model. While effective for generation, this two-stage paradigm is inefficient. In contrast, our method directly distills teacher CFG prediction through MeanFlow distillation, avoiding two-stage training and improving efficiency.

**S3Diff (Zhang et al., 2024).** S3Diff introduces an online negative sample generation strategy to align low-quality concepts with negative prompts, enabling CFG at inference to improve visual quality. However, this requires applying CFG during the inference time of the student model, effectively doubling the NFE. By contrast, our approach utilize the negative prompt enhanced teacher's CFG prediction as the supervision signal during training, allowing genuine 1 NFE inference.

## F    MORE ABLATION STUDY RESULTS

**Ratio of $t \neq s$.** We study the effect of varying the ratio of $t \neq s$ on RealLQ250 in Table 4. Empirically, a ratio of 0.5 yields the best results, which is lower than the 0.75 used in original MeanFlow (Geng et al., 2025). This difference arises because our distillation setting already captures the instantaneous velocity field, allowing greater focus on learning the shortcut.

**Visual comparison of CFG strategies.** In Fig. 6, we provide a visual comparison from the ablation study of our proposed CFG strategy. All variants perform 1-step sampling. Our method delivers the best restoration quality, free of artifacts and with the most detailed textures.

Table 4: Ablation studies for hyperparameter ratio $r$.

| $r$ | NIQE ↓ | MUSIQ ↑ | MANIQA ↑ | CLIPIQA ↑ |
|-----|--------|---------|----------|-----------|
| 0 | 5.2529 | 67.23 | 0.6283 | 0.6819 |
| 0.25 | 5.2311 | 67.58 | 0.6301 | 0.6863 |
| 0.5 | 5.2421 | **67.95** | **0.6389** | **0.6968** |
| 0.75 | **5.1046** | 67.76 | 0.6358 | 0.6895 |

|  LR  |  $z_1 - z_0$  |  Original MeanFlow CFG  |  Ours  |

Figure 6: Visual comparison from ablation study of our CFG strategy.

## G  QUALITATIVE COMPARISON ACROSS DIFFERENT SAMPLING STEP AND COMPARISON WITH THE TEACHER MODEL

To better illustrate the effectiveness of our method, we present qualitative comparisons between MFSR (with 1/2/4 steps) and the teacher model in Fig. 7. Performing only a single inference step with DiT4SR results in pronounced artifacts and distortions. The first three rows compare super-resolution results across different sampling steps of our student model (1/2/4 steps) against the teacher model. Our one-step restoration occasionally introduces reconstruction errors; for example, in the first row, the reflection in the water is incorrectly reconstructed as buildings. In contrast, our two-step and four-step variants effectively correct this issue, producing realistic water ripples and reflection. In the second row, our one-step restoration fails to remove background degradations around the cat's ear, whereas two-step and four-step restoration successfully remove these artifacts and produce sharper, more realistic fur details compared to the teacher. The third row shows an image containing text: the one-step model distorts the letter $M$, while two-step and four-step models accurately reconstruct the character. These examples demonstrate that increasing the number of sampling steps improves restoration quality, offering a flexible trade-off between efficiency and SR quality.

The fourth to sixth rows highlight cases where our method surpasses the teacher model. Specifically, our approach yields sharper and more natural reconstructed leaves (while the teacher outputs blurry textures), more realistic wall patterns, and a better removal of excessive blur.

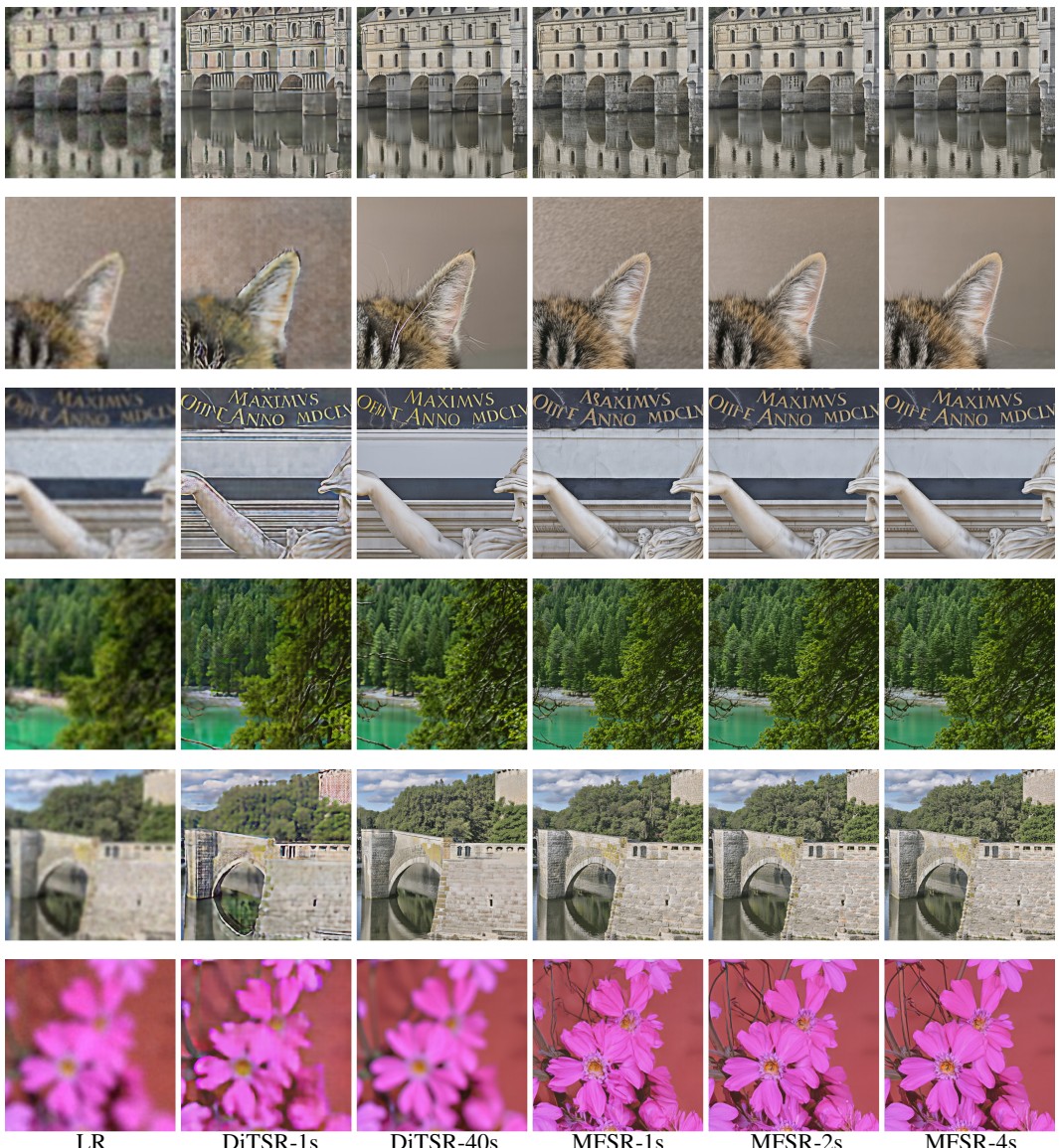

LR            DiTSR-1s         DiTSR-40s         MFSR-1s          MFSR-2s          MFSR-4s

Figure 7: Qualitative comparison across different sampling steps and our teacher model, DiT4SR. The number of diffusion inference steps is indicated by 's'. Please zoom in for a better view.

Overall, these results indicate that our distilled student model achieves restoration quality on par with, or even superior to the teacher model and is much more efficient.

## H    COMPARISON WITH MULTI-STEP DIFFUSION-BASED METHODS

In Table 5, we present a quantitative comparison with representative multi-step diffusion-based methods on the DRealSR and RealSR datasets. The competing methods include StableSR (Wang et al., 2023), DiffBIR (Lin et al., 2023), SeeSR (Wu et al., 2024b), SUPIR (Yu et al., 2024), PASD (Yang et al., 2023), ResShift (Yue et al., 2024), and the teacher model DiT4SR (Duan et al., 2025). Our approach demonstrates best or competitive performance while being much less denoising steps than these multi-step counterparts.

Table 5: Quantitative comparison with state-of-the-art multi-step methods on real-world benchmarks. The number of diffusion inference steps is indicated by 's'. The best and second best results of each metric are highlighted in red and blue, respectively.

| Datasets | Method | PSNR ↑ | SSIM ↑ | LPIPS ↓ | DISTS ↓ | FID ↓ | NIQE ↓ | MUSIQ ↑ | MANIQA ↑ | CLIPIQA ↑ |
|---|---|---|---|---|---|---|---|---|---|---|
| DRealSR | StableSR-200s | 28.04 | 0.7454 | 0.3279 | 0.2272 | 144.15 | 6.5999 | 58.53 | 0.5603 | 0.6250 |
| | DiffBIR-50s | 25.93 | 0.6525 | 0.4518 | 0.2761 | 177.04 | 6.2324 | 65.66 | 0.6296 | 0.6860 |
| | SeeSR-50s | 28.14 | 0.7712 | 0.3141 | 0.2297 | 146.95 | 6.4632 | 64.74 | 0.6022 | 0.6893 |
| | SUPIR-50s | 25.09 | 0.6460 | 0.4243 | 0.2795 | 169.48 | 7.3918 | 58.79 | 0.5471 | 0.6749 |
| | DiT4SR-40s | 25.69 | 0.6802 | 0.3644 | 0.2442 | 156.95 | 6.6407 | 64.39 | 0.6230 | 0.6561 |
| | PASD-20s | 27.79 | 0.7495 | 0.3579 | 0.2524 | 171.03 | 6.7661 | 63.23 | 0.5919 | 0.6242 |
| | ResShift-15s | 28.69 | 0.7874 | 0.3525 | 0.2541 | 176.77 | 7.8762 | 52.40 | 0.4756 | 0.5413 |
| | MFSR-1s | 24.15 | 0.6423 | 0.3660 | 0.2379 | 143.12 | 6.0241 | 64.47 | 0.6148 | 0.7171 |
| | MFSR-2s | 24.29 | 0.6455 | 0.3689 | 0.2333 | 139.56 | 6.2711 | 64.45 | 0.6354 | 0.7023 |
| RealSR | StableSR-200s | 24.62 | 0.7041 | 0.3070 | 0.2156 | 128.54 | 5.7817 | 65.48 | 0.6223 | 0.6198 |
| | DiffBIR-50s | 24.24 | 0.6650 | 0.3469 | 0.2300 | 134.56 | 5.4932 | 68.35 | 0.6544 | 0.6961 |
| | SeeSR-50s | 25.21 | 0.7216 | 0.3003 | 0.2218 | 125.10 | 5.3978 | 69.69 | 0.6443 | 0.6671 |
| | DiT4SR-40s | 23.50 | 0.6683 | 0.3173 | 0.2239 | 118.94 | 6.0077 | 67.85 | 0.6587 | 0.6398 |
| | SUPIR-50s | 23.65 | 0.6620 | 0.3541 | 0.2488 | 130.38 | 6.1099 | 62.09 | 0.5780 | 0.6707 |
| | PASD-20s | 25.68 | 0.7273 | 0.3144 | 0.2304 | 134.18 | 5.7616 | 68.33 | 0.6323 | 0.5783 |
| | ResShift-15s | 26.39 | 0.7567 | 0.3158 | 0.2432 | 149.59 | 6.8746 | 60.22 | 0.5419 | 0.5496 |
| | MFSR-1s | 21.51 | 0.6347 | 0.3158 | 0.2295 | 110.14 | 5.2421 | 67.95 | 0.6389 | 0.6968 |
| | MFSR-2s | 21.75 | 0.6494 | 0.2999 | 0.2222 | 107.87 | 5.5980 | 67.45 | 0.6560 | 0.6705 |

# I    LR Images in User Study

Fig. 8 shows the thumbnail of LR images used in the user study.

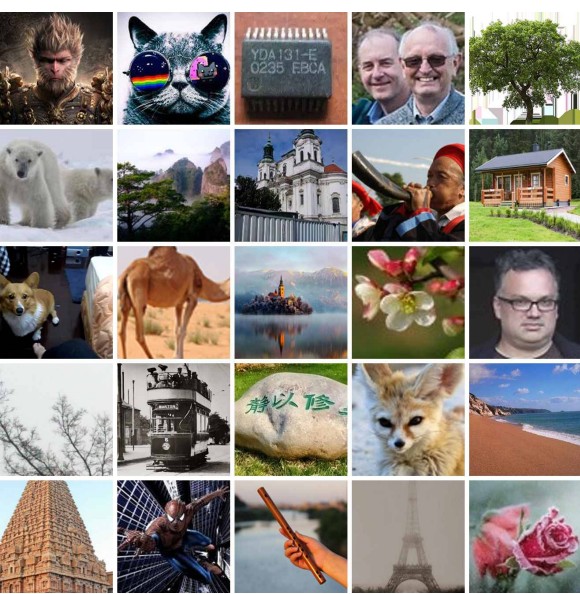

Figure 8: The LR images used in user study.

# J    More Visual Comparisons

In Fig. 9 and Fig. 10, we provide additional visual comparisons with other state-of-the-art one-step methods, further demonstrating the robust restoration ability of MFSR and the superior quality of its results.

In addition, Fig. 11 presents examples of super-resolution on AI-Generated Content (AIGC), and Fig. 12 shows an example of old photo restoration. These results achieve visually pleasing effects, highlighting strong practical value of our method in real-world applications.

## K    USE OF LARGE LANGUAGE MODELS

Large language models were employed solely to refine language and correct grammar in the manuscript. They played no role in the conception or design of the methodology, experiments, or data analysis. The authors independently verified and validated all technical content, results, and conclusions.

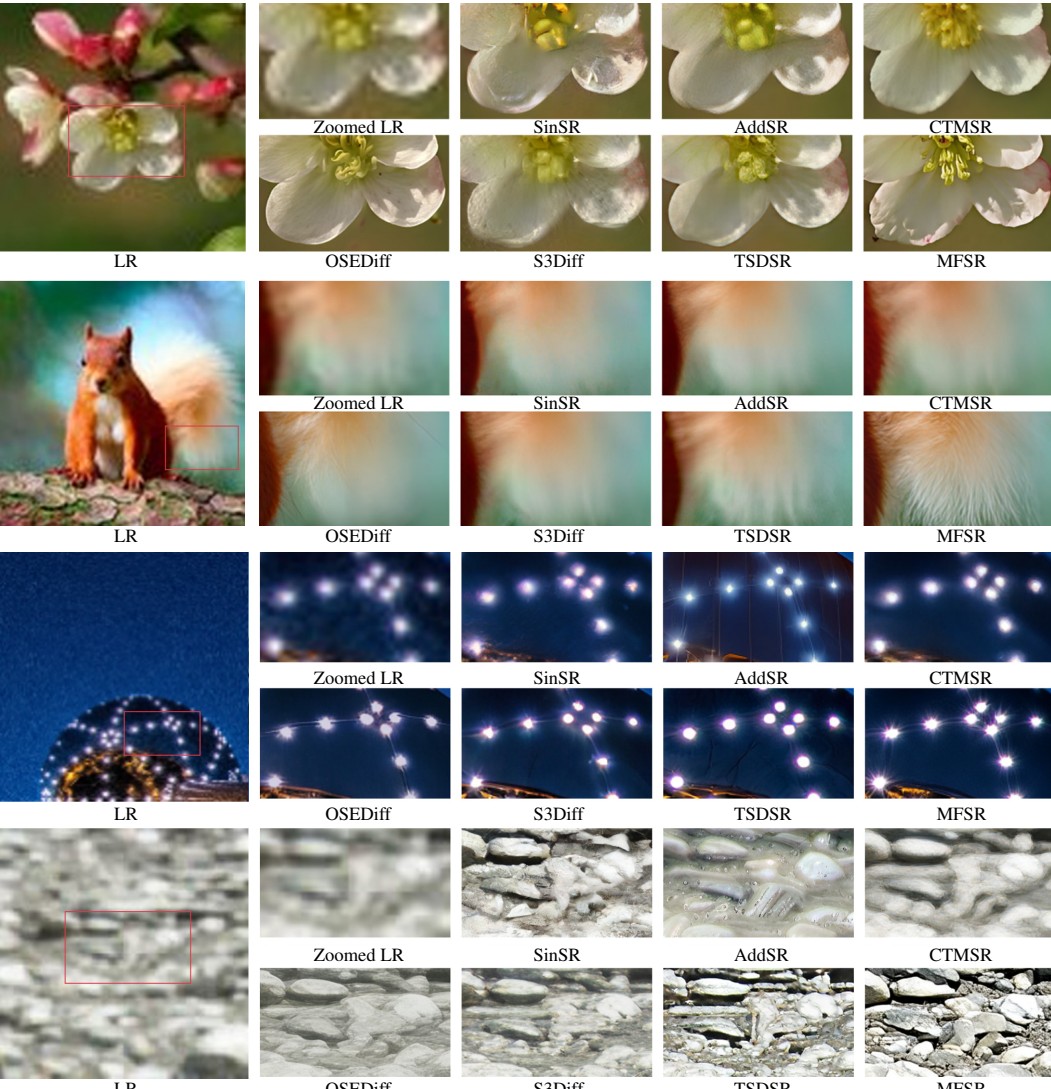

Figure 9: Qualitative comparison with state-of-the-art methods. All methods perform 1-step inference. Please zoom in for a better view.

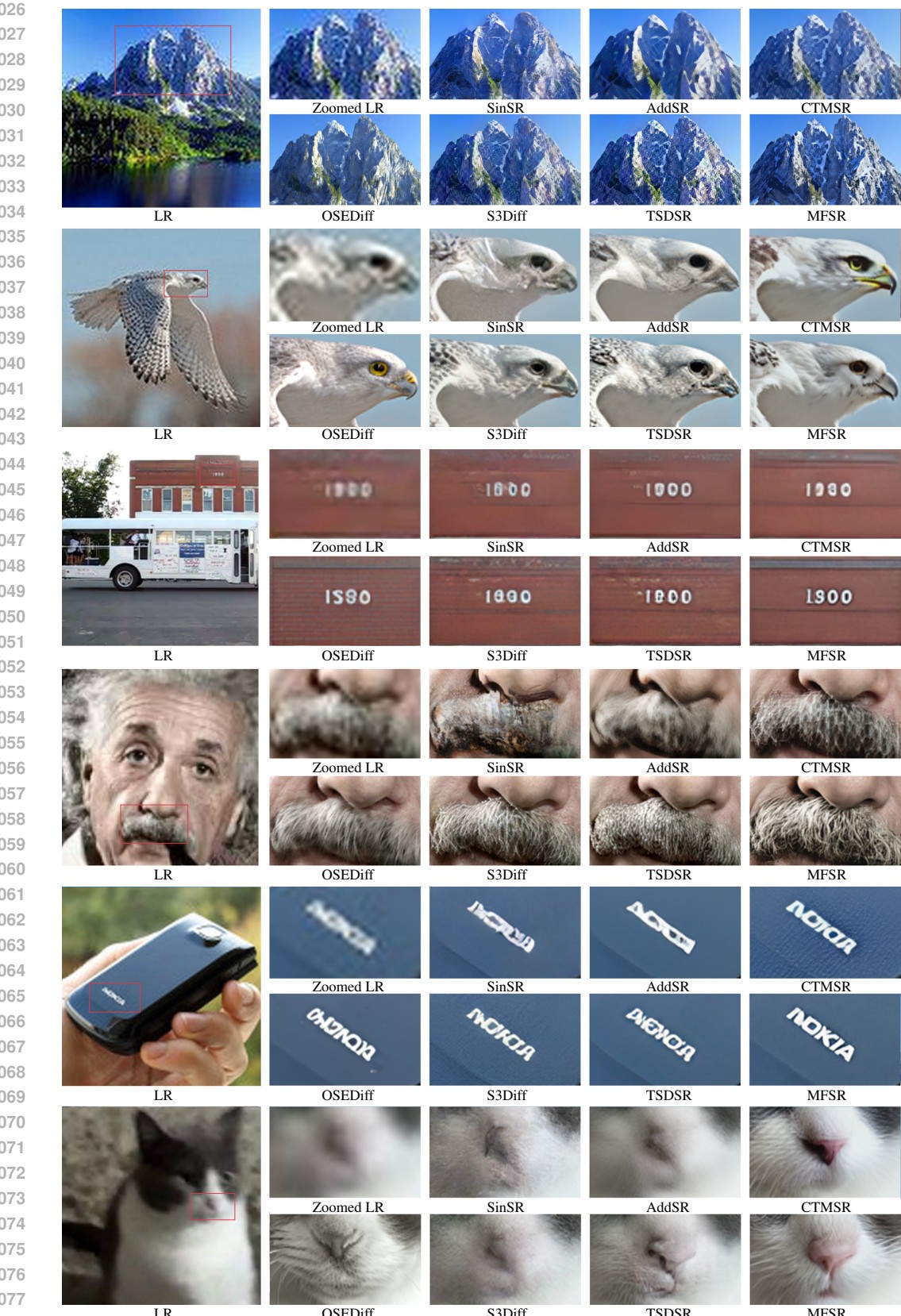

Figure 10: Qualitative comparison with state-of-the-art methods. All methods perform 1-step inference. Please zoom in for a better view.

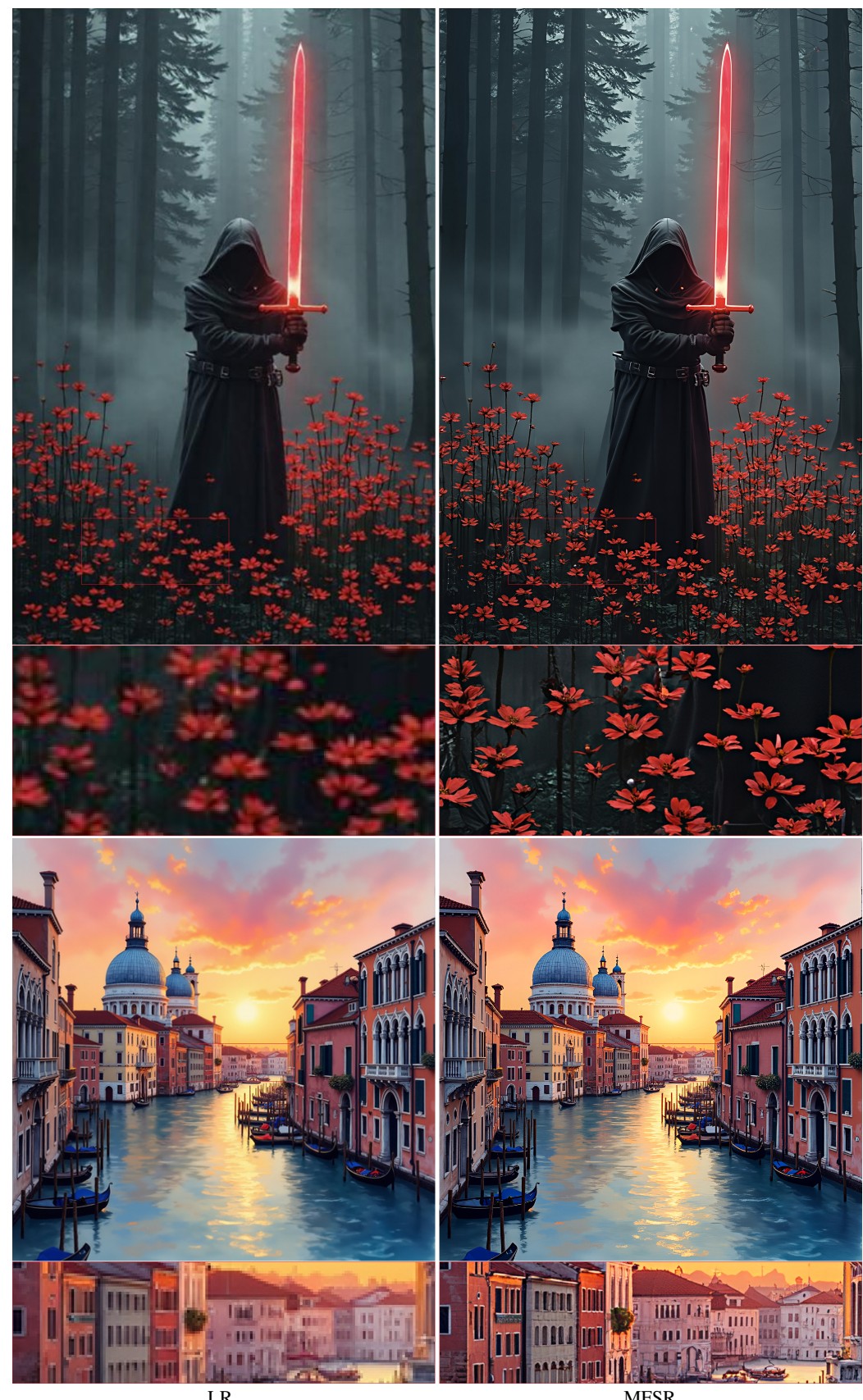

Figure 11: 4× SR results on AI-Generated Content using 3-step sampling. Please zoom in for a better view.

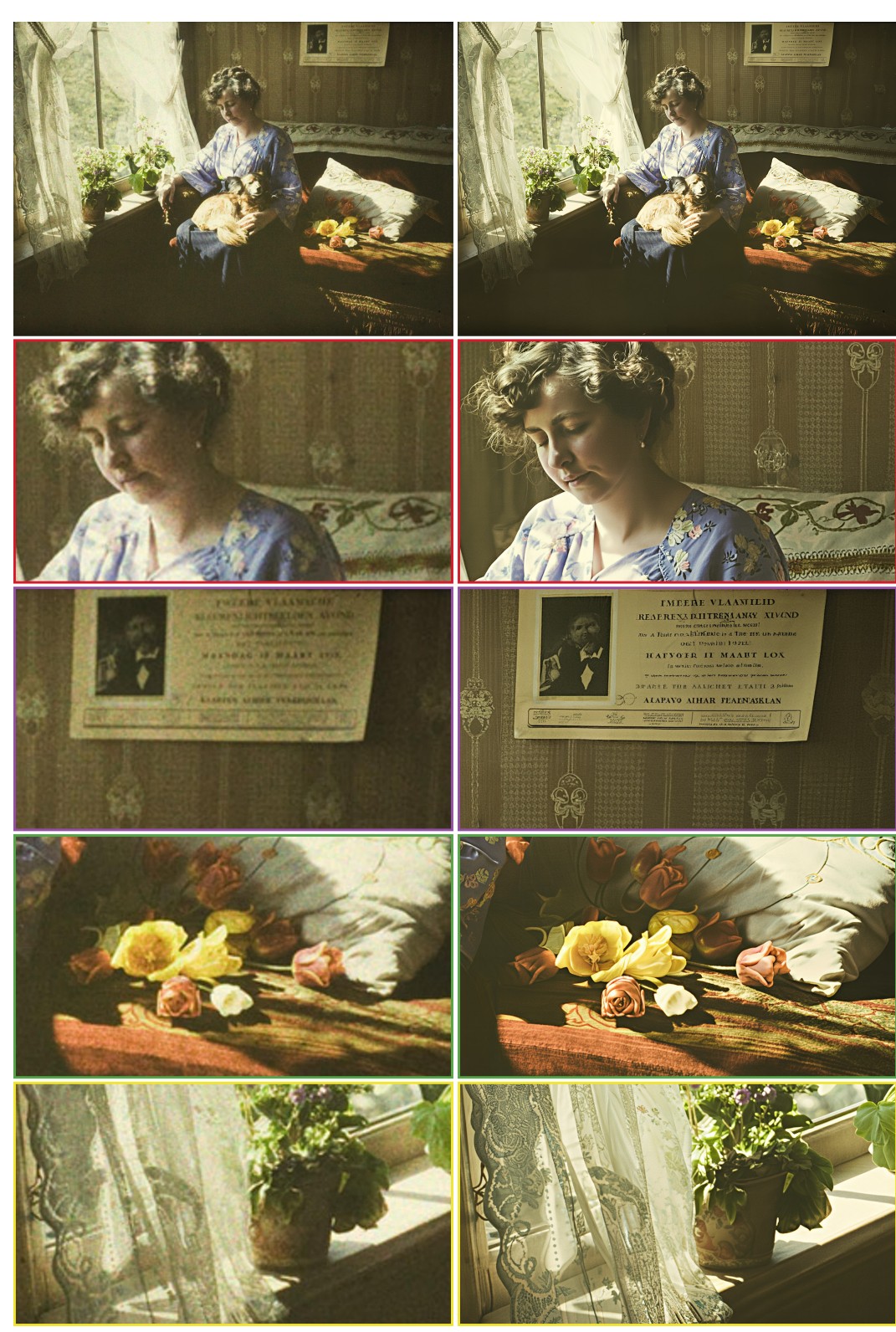

LR            MFSR

Figure 12: Result of old photo restoration using 3-step sampling. Please zoom in for a better view.

