# OpenReview forum: "MFSR: MeanFlow Distillation for One Step Real-World Image Super Resolution"
_ICLR.cc/2026/Conference — ICLR 2026 Conference Withdrawn Submission_

### Official Review · Reviewer_6iut · 2025-10-16

**Soundness:** 3
**Presentation:** 2
**Contribution:** 3
**Rating:** 4
**Confidence:** 3

**Summary:**

This paper presents MFSR, which brings MeanFlow to real image super resolution. It can restore images in one step or in a few steps. The key idea is a new CFG-based MeanFlow distillation method. It uses a strong teacher model DiT4SR to guide training, giving better supervision than the original MeanFlow CFG approach.

**Strengths:**

(i) The authors reformulate classifier-free guidance for MeanFlow [4] to a distillation regime by leveraging a pretrained teacher during training. This CFG-based MeanFlow distillation yields stronger supervision than the original MeanFlow CFG, improving one and few step restoration quality.

(ii) The approach is scaled to a DiT architecture [1,2,3], using a pretrained DiT4SR [3] both as the teacher and for student initialization. This scaling to relatively big models is nontrivial and demonstrates the method’s ability to work well with larger models, where it was not clear if the method would work.

(iii) Extensive empirical evaluations demonstrate that MFSR achieves competitive performance compared to existing single step super resolution approaches.

[1] https://arxiv.org/abs/2403.03206 Scaling Rectified Flow Transformers for High-Resolution Image Synthesis

[2] https://arxiv.org/abs/2212.09748 Scalable Diffusion Models with Transformers

[3] https://arxiv.org/abs/2503.23580 DiT4SR: Taming Diffusion Transformer for Real-World Image Super-Resolution

[4] https://arxiv.org/abs/2505.13447 Mean Flows for One-step Generative Modeling

**Weaknesses:**

(i) Using MeanFlow with distillation and CFG seems like a natural next step. While useful, this idea is not very new or surprising. The paper feels more like an engineering improvement than a theoretical one.

(ii) The authors say metrics don’t reflect real-world SR well and use a user study to support this. However, the human evaluation compares only a few methods, so the results are not very strong. A larger study with more methods would help. Since the paper focuses on practical SR, it should compare with other strong non-diffusion models like SwinIR [5], Real-ESRGAN [6], and others [7,8,9].

(iii) It’s not clear if the good results come from the method or just from using a large DiT model. The paper should compare with other distillation methods using the same setup and also show compute cost and ablations.

[5] https://arxiv.org/abs/2108.10257 SwinIR: Image Restoration Using Swin Transformer

[6] https://arxiv.org/abs/2107.10833 Real-ESRGAN: Training Real-World Blind Super-Resolution with Pure Synthetic Data

[7] https://arxiv.org/abs/2503.13358 One-Step Residual Shifting Diffusion for Image Super-Resolution via Distillation

[8] https://arxiv.org/abs/2411.13383 Adversarial Diffusion Compression for Real-World Image Super-Resolution

[9] https://arxiv.org/abs/2103.14006 Designing a Practical Degradation Model for Deep Blind Image Super-Resolution

**Questions:**

(i) I couldn’t understand what the authors wanted to show in Appendix C. It’s not clearly mentioned in the main text. Could the authors explain its purpose and whether it supports any main claim?

(ii) Why do the authors use a text extractor on the HR image during training, but on the LR image during inference? Does this difference create any bias or mismatch between training and testing?

(iii) Since the student never sees the real HR image during distillation, how can it perform better than the teacher on non-reference metrics (as in Table 5)? Can the authors give some explanation or intuition for this?

---

### Official Review · Reviewer_d2tE · 2025-10-27

**Soundness:** 2
**Presentation:** 3
**Contribution:** 2
**Rating:** 4
**Confidence:** 3

**Summary:**

The authors propose MFSR, a one-/few-step real-world super-resolution method. It builds directly on the recently introduced Mean Flow framework, using it to distill the multi-step DiT4SR super-resolution teacher into an efficient student model. Rather than estimating the instantaneous velocity with a one-sample estimator, the student is supervised using the instantaneous velocity predicted by the teacher. Moreover, the authors enhance this supervision by applying classifier-free guidance - including strong guided and negative-prompt variants - directly to the teacher’s velocity during distillation, further improving the perceptual quality of the distilled student.

**Strengths:**

1. The authors propose a new view on Mean Flow, not as a training from scratch approach, but as a distillation approach using the instantaneous velocity predicted by the teacher instead of a one-sample estimator.

2. Good quality of the resulting model.

**Weaknesses:**

1. **Limited novelty and lack of SR-specific contribution.**
While the method differs from the previously proposed Mean Flow approach by using instantaneous velocity provided by the teacher model, it introduces no super-resolution-specific modeling or insight. As a result, it does not meaningfully address the distinct challenges of the SR domain, and the contribution reads more as an engineering transfer rather than SR-oriented innovation.

2. **Insufficient analysis and insight.**
The paper does not analyze why previously proposed one-step/inference-efficient SR methods fail or in which specific aspects the proposed approach succeeds. There is no principled discussion of what challenge is being solved, or what underlying mechanism makes MFSR superior in certain cases. The work remains largely observational (leaderboard-style) rather than explanatory or diagnostic.

3. **Unfair and under-discussed comparison setup.**
Many of the baseline SR methods being compared against are themselves distilled from different teacher models under different training conditions. This heterogeneity is not acknowledged or controlled for. Moreover, the authors train their own student using yet another teacher (DiT4SR) that is not aligned with the baselines. As a result, the experimental comparisons lack a fair common-teacher reference, making it unclear whether improvements stem from the proposed method or simply from a stronger teacher.

**Questions:**

1. **Choice of teacher model.**
Why did the authors specifically choose DiT4SR as the teacher for distillation?

2. **Prompt inconsistency between training and inference.**
During training, prompts are extracted from the high-resolution (HR) image, whereas during inference, they are extracted from the low-resolution (LR) input. What is the rationale behind introducing this train–test mismatch? Wouldn't it be better to use the LR input on the train?

---

### Official Review · Reviewer_eT5Y · 2025-10-31

**Soundness:** 2
**Presentation:** 3
**Contribution:** 3
**Rating:** 4
**Confidence:** 4

**Summary:**

The paper introduces MeanFlows for Super-Resolution (MFSR), a one-step or few-step image super-resolution (ISR) model for Real-World ISR task. MFSR aims to improve the efficiency of flow-based diffusion models by reducing sampling steps. The approach uses a distillation framework to transfer knowledge from a powerful multi-step teacher model (e.g. DiT4SR) into a faster student model, while also introducing improvements to the Classifier-Free Guidance (CFG) formulation. The proposed method shows promise in restoring high-fidelity images with fewer computational steps, outperforming or matching existing methods in both synthetic and real-world image benchmarks.

**Strengths:**

1. The proposed method, MFSR, reduces computational cost while maintaining quality. The use of the MeanFlow distillation strategy enables high-quality one-step restoration and retains flexibility for optional few-step refinement.

2. The idea of leveraging a pre-trained teacher model to guide the student model in distilling knowledge is well-executed. This reduces convergence time and improves the quality of restoration, making the approach more practical for real-world applications.

3. The modification of the Classifier-Free Guidance (CFG) in the distillation process enhances the model’s performance, particularly in preserving fine details and improving perceptual quality.

**Weaknesses:**

1. The main difference between MFSR and MeanFlow lies in the use of teacher CFG to improve instantaneous velocity. Therefore, the technical contribution appears incremental rather than fundamentally novel.

2. The quantitative results reported show limited competitiveness compared with other methods. MFSR fails to achieve state-of-the-art performance on all FR metrics. While I understand that PSNR and SSIM are less meaningful in Real-ISR, LPIPS and DISTS remain important reference indicators. Among the four real-world datasets, MFSR only outperforms others on the NR metric MANIQA. Although MFSR supports two-step sampling, it still falls short of TSDSR on most metrics.

**Questions:**

1. In Section 4.3, the authors claim that the naive choice of instantaneous velocity $z_1 - z_0$ leads to inferior reconstruction results. I guess this is caused by a mismatch between the noise and the data, resulting in an incorrect estimation of the instantaneous velocity. The authors should include additional experiments using the teacher-predicted $z_0$ to compute the instantaneous velocity in order to clarify this point.

2. Could the authors provide a comparison of runtime efficiency and parameter count with other methods?

---

### Official Review · Reviewer_LiJf · 2025-11-03

**Soundness:** 3
**Presentation:** 3
**Contribution:** 3
**Rating:** 4
**Confidence:** 3

**Summary:**

This paper proposes MFSR, a distillation framework that adapts MeanFlow to real-world image super-resolution by distilling a multi-step DiT4SR teacher into a one-step student model while maintaining the flexibility for few-step sampling. The key contribution is an improved CFG-based distillation strategy that uses the teacher's CFG-enhanced prediction as instantaneous velocity. While the paper addresses the relevant problem of reducing inference cost in Real-ISR, significant concerns remain regarding experimental validation and methodological justification. Most critically, MFSR consistently underperforms baseline methods across multiple fidelity metrics (PSNR, SSIM, LPIPS, DISTS), and the paper does not adequately address this degradation or provide sufficient evidence for the claimed superiority of the proposed approach.

**Strengths:**

1.	Well-motivated problem: The paper addresses a practical challenge in real-world super-resolution - reducing inference cost while maintaining quality. The motivation for adapting MeanFlow to SR distillation is clear and reasonable.
2.	Flexible inference scheme: Unlike existing one-step SR methods, MFSR preserves the ability to perform few-step sampling, providing a controllable trade-off between speed and quality. This is a valuable feature for practical deployment.
3.	Simplified training pipeline: The method uses only the MeanFlow distillation loss computed in latent space without backpropagating through encoder/decoder, which improves training efficiency compared to methods like OSEDiff and TSDSR.

**Weaknesses:**

1.	Inconsistent and concerning quantitative results:
(1) MFSR shows significantly lower PSNR/SSIM across all benchmarks (e.g., 21.25 vs 24.50 PSNR on DIV2K-Val for SinSR). While the authors mention perception-distortion tradeoff, the gap is substantial.
(2) The paper dismisses these metrics as "misaligned with human perception" but still reports them extensively, creating confusion about which metrics to trust.
(3) On some metrics (LPIPS, DISTS), MFSR performs worse than several baselines, contradicting the claim of superior perceptual quality.
2.	Why does MFSR have such low PSNR/SSIM compared to baselines? At what point does the perception-distortion tradeoff become concerning for practical applications?
3.	Questionable user study design: The instruction to select results that "best balance realism and fidelity" is subjective and may introduce bias toward methods favoring perceptual quality at the expense of fidelity. This potentially skews the evaluation in favor of MFSR's characteristics rather than providing an objective assessment of quality.
4.	Missing details about prompt extraction method, negative prompt selection, and time step sampling distribution.
5.	The statement "delivers results on par with or even better than multi-step teachers" is not well-supported by the quantitative results shown.
6.	The paper does not demonstrate why MeanFlow-based distillation is superior to directly distilling the teacher into a one-step or few-step instantaneous velocity model (standard Rectified Flow). What if the same teacher model (DiT4SR) were distilled using trajectory-based methods that predict instantaneous velocity rather than average velocity? This comparison is essential to validate the claimed advantage of MeanFlow.
7.	The paper should compare with more similar recent flow-based one-step SR methods. For example, "FlowSR: One Diffusion Step to Real-World Super-Resolution via Flow Trajectory Distillation" also distills multi-step flow matching models and represents a highly relevant baseline.

**Questions:**

Please see the weaknesses above.

---

### Note · Authors · 2025-11-14

I have read and agree with the venue's withdrawal policy on behalf of myself and my co-authors.